# Assessing Gut Microbiota in an Infant with Congenital Propionic Acidemia before and after Probiotic Supplementation

**DOI:** 10.3390/microorganisms9122599

**Published:** 2021-12-16

**Authors:** Andrea Bordugo, Elisa Salvetti, Giulia Rodella, Michele Piazza, Alice Dianin, Angela Amoruso, Giorgio Piacentini, Marco Pane, Sandra Torriani, Nicola Vitulo, Giovanna E. Felis

**Affiliations:** 1Inherited Metabolic Disease Unit, Pediatric Clinic C, Azienda Ospedaliera Università Integrata, Piazzale Stefani 1, 37126 Verona, Italy; giulia.rodella@univr.it (G.R.); alice.dianin@aovr.veneto.it (A.D.); 2Department of Biotechnology, University of Verona, Strada le Grazie 15, 37134 Verona, Italy; sandra.torriani@univr.it (S.T.); nicola.vitulo@univr.it (N.V.); giovanna.felis@univr.it (G.E.F.); 3Department of Surgery, Dentistry, Paediatrics and Gynaecology, University of Verona, Piazzale Stefani 1, 37126 Verona, Italy; michele.piazza@univr.it (M.P.); giorgio.piacentini@univr.it (G.P.); 4Probiotical Research Srl, Via Enrico Mattei 3, 28100 Novara, Italy; a.amoruso@probiotical.com (A.A.); m.pane@probiotical.com (M.P.)

**Keywords:** propionic acidemia, propionic acid, metronidazole, inherited metabolic diseases, metabarcoding, metagenomics, organic acidurias, diet, genetic, electronic nose, VOC, gut microbiota, *Bacteroides fragilis*, probiotic supplementation, bifidobacteria, *Bifidobacterium breve*, *Ruminococcus gnavus*

## Abstract

Propionic Acidemia (PA) is a rare inherited metabolic disorder caused by the enzymatic block of propionyl-CoA carboxylase with the consequent accumulation of propionic acid, which is toxic for the brain and cardiac cells. Since a considerable amount of propionate is produced by intestinal bacteria, interest arose in the attempt to reduce propionate-producing bacteria through a monthly antibiotic treatment of metronidazole. In the present study, we investigated the gut microbiota structure of an infant diagnosed at 4 days of life through Expanded Newborn Screening (NBS) and treated the child following international guidelines with a special low-protein diet, specific medications and strict biochemical monitoring. Microbiota composition was assessed during the first month of life, and the presence of *Bacteroides fragilis,* known to be associated with propionate production, was effectively decreased by metronidazole treatment. After five antibiotic therapy cycles, at 4 months of age, the infant was supplemented with a daily mixture of three bifidobacterial strains, known not to be propionate producers. The supplementation increased the population of bifidobacteria, with *Bifidobacterium breve* as the dominating species; *Ruminococcus gnavus*, an acetate and formate producer, was also identified. Metabarcoding analysis, compared with low coverage whole metagenome sequencing, proved to capture all the microbial biodiversity and could be the elected tool for fast and cost-effective monitoring protocols to be implemented in the follow up of rare metabolic disorders such as PA. Data obtained could be a possible starting point to set up tailored microbiota modification treatment studies in the attempt to improve the quality of life of people affected by propionic acidemia.

## 1. Introduction

Propionic acidemia (PA) is a rare inherited autosomal recessive metabolic disorder with a reported incidence of 1:100,000 newborns in Europe and 1:242,741 in the United States (OMIM 606054; https://omim.org/entry/606054, last accessed on 21 May 2021). It is caused by a mutation in the genes PCCA or PCCB encoding the mitochondrial enzyme propionyl-CoA carboxylase. These two genes are involved in the isoleucine (ILE), valine (VAL), methionine (MET), and threonine (THR) catabolic pathways as well as in the catabolism of odd-chain fatty acids and the cholesterol side chain, converting them to methylmalonyl-CoA, which is then metabolized to succinyl-CoA (further oxidized in the citric acid cycle for ATP production) [1].

Defective function of these genes results in the accumulation of propionic acid metabolites, and dysfunction in the respiratory chain and urea cycle pathways [2].

Neonatal onset of PA is characterized by metabolic acidosis with an increased anion gap, ketonuria, hypoglycemia, hyperammonemia, and cytopenia. The clinical features include poor feeding, episodic vomiting, fatigue, followed by developmental retardation, intolerance to protein, lethargy, seizures, coma, and death if not treated. Late childhood cardiac involvement is one of the major issues.

Expanded newborn screening (NBS) detects high propionylcarnitine levels through tandem mass spectrometry (MS/MS). PA diagnosis in newborns is then confirmed by high levels of methylcitric acid (MCA) on dried blood samples (DBS) or plasma, propionic acid in urine, and bi-allelic gene mutations [3]. Several newborns became symptomatic before their NBS results were available.

On 1 January 2014, the Veneto Region (Italy) introduced an expanded neonatal screening program to facilitate the early identification of inherited metabolic disorders (IMD) (including PA) (Regional Law 1308/13), thus allowing prompt therapy implementation, preventing much of the symptomatology and slowing down the disease progression. In 2016 a National Law (n. 167/2016) made NBS mandatory in all Italian regions.

Following the diagnosis, there is no wide consensus on how to treat patients with suspected or confirmed PA, although some guidelines and recommendations are available [4].

The standard therapy for PA management mainly includes a low-protein diet, the use of antibiotics to reduce gut propionate production from intestinal microbiota, L-carnitine, precursor-free amino acid and/or isoleucine/valine supplementation, as well as vitamin and mineral supplementation [4,5].

Among antibiotics, metronidazole has been proven useful to reduce the production of propionyl-CoA derived from the anaerobic bacterial fermentation of carbohydrates in the gut, which may account for a large proportion of total body propionate [4]. In order to avoid the development of drug-resistant colonies and reduce the therapy side-effects (which may include QTc prolongation and pancreatitis) [6], metronidazole is administered for 1–2 weeks alternating with 2–3 weeks off. Furthermore, probiotic supplementation (avoiding those containing propionic acid producers) has been suggested as this could lead to a clinical improvement by replacing the propionate-producing populations in the gut as well as restoring the normal gut microbiome composition [4,7].

In the present study, we described the results of an investigation on the gut microbiota composition of an infant, born in May 2020, early diagnosed with PA, who immediately entered the standard protocols of therapy (low-protein diet and metronidazole) and PA-related biochemical monitoring, including urine analysis with a biomimetic device. Microbiota composition was assessed during the first month of life, filling the gap of knowledge on the propionate-producers identity. After five metronidazole treatments, at the age of 4 months, the infant also received a supplementation of the standard diet with a well-characterized probiotic bifidobacteria (low dosage mixture of three strains produced on purpose), and the gut microbiota structure were assessed as before.

Molecular analyses were performed with both metabarcoding and low coverage whole metagenome sequencing to be able to capture all the microbial biodiversity and determine the election tool for further analyses.

## 2. Materials and Methods

### 2.1. Patient Description

The patient was born at term after an uneventful pregnancy. He started breastfeeding soon after birth. After 48 h, he showed feeding intolerance and vomiting. Intravenous glucose was started for mild metabolic acidosis (BE-10 and ammonia levels 120 umol/L). At 4 days of age, NBS results showed high propionylcarnitine levels (C3); further, second-tier testing revealed high levels of methylcitric acid (MCA) with normal levels of methylmalonic acid (MMA) and homocysteine (HCY). Plasma C3, organic acid analysis, and genetic testing confirmed the diagnosis. A homozygous mutation c.1118 T > A (p.Met373Lys) in the PCCA gene was found, classified as pathogenic [8,9]. Immediately after the diagnosis, the patient started a low-protein diet, carnitine (200 mg/kg/day), carglumic acid (100 mg/kg/day), and metronidazole ¼ weeks (15 mg/kg/day).

### 2.2. Clinical and Biochemical Disease Specific Follow-Up

Regular visits were performed at the Clinical Department (Inherited Metabolic Disease Unit, Pediatric Clinic C, Azienda Ospedaliera Universitaria Integrata, AOUI Verona, Italy); telemedicine technology was applied for a strict follow-up. Clinical and biochemical data were collected every 2–3 months of life (B1–B5, Figure 1; a further analysis was carried out in January 2021, at 8 months of age, not mapped in Figure 1), and they included: weight/growth status, neurological evaluation, and other biochemical parameters such as blood gas analysis and ammonia, biochemical and hematological profiling. In addition, disease-related metabolic profiles were monitored: organic acids and acylcarnitines were checked through tandem mass spectrometry (MS/MS), either in plasma (both organic acids and acylcarnitines) and via dried blood spot sampling (DBS, only acylcarnitines), while methylcitrate was determined in DBS by tandem mass spectrometry combined with ultra performance liquid chromatography (UPLC-MS/MS).

### 2.3. Diet Monitoring

Since the confirmation of the diagnosis, dietary treatment began according to the recommendations for PA [4]. The patient started a specific low-protein diet to reduce the intake of precursors of propionyl-CoA on one side, but at the same time, to achieve the safe level for protein requirements declared by FAO (FAO/WHO/UNU 2007) [10]. An emergency regimen was also adopted for intercurrent illness in order to stop protein intake for the first 24 h but ensuring adequate calories with a protein-free nutritional formula rich in maltodextrins and fat.

According to the patient’s diet prescription, a controlled amount of maternal milk was used: the maternal milk was collected, stored in a domestic refrigerator at 3–4 °C and used within 24 h of collection. Infant formula and protein-free formula were prepared following the WHO recommendations on powder milk preparation. Diet was monitored through food diaries according to the clinical follow-up, and the nutritional analysis was performed with Metadieta^®^ software (version PRO 3.4, Meteda, S. Benedetto del Tronto, Italy). Diet changes occurred frequently considering the patient requirements, growth, and biochemical analysis.

### 2.4. Urine Analysis with the Electronic Nose

Urine samples (U1–U4) were collected in a urine bag according to the timeline reported in Figure 1, then frozen at −20 °C until analysis.

Urine analyses were performed with a commercial electronic nose (Cyranose 320; Sensigent, Pasadena, CA, USA) on three batches of every individual urination. The electronic nose contained a nanocomposite array comprising 32 polymer sensors, each having a different coating. When volatile organic compounds (VOCs) in samples were exposed to the nanocomposite array, the polymers swelled, inducing a change in their electrical resistance. These changes were recorded and collected in an onboard database to generate a distribution graph (smell-print) that described the VOCs mixture and which could be used for pattern-recognition algorithms. VOCs were compared before and after the administration of metronidazole and after the probiotic formula.

To limit potential confounding factors, three batches per urine sample were processed on the same day. Samples were thawed at 37 °C to increase the concentration of VOCs in the headspaces; furthermore, after each measurement, sensors were purged for 90 s to remove any residue. Three measurements were conducted for each sample. The values of each probe were standardized, and all the subsequent statistical analyses were performed on the mean values of the standardized replicates.

### 2.5. Probiotic Supplementation

The probiotic product was administered for three weeks during month 4 following the 5th one-week treatment with metronidazole (Figure 1). The patient received one capsule (300 mg) every day with a total of 3 × 10^9^ live cells (CFU\AFU) of the probiotic mixture containing *Bifidobacterium longum* 04 (DSM 23233), *Bifidobacterium bifidum* BB01 (DSM 22892), and *Bifidobacterium breve* BR03 (DSM 16604). These probiotic strains are commercially available and are extensively used for food supplement formulations. The study materials were analyzed by Biolab Research S.r.l., Novara, Italy, via flow cytometry (ISO 19344:2015 IDF 232:2015) and the plate count method (Biolab Research Method 014-06) to confirm the target cell count.

The probiotic supplementation was included in a compassionate off-label use study which was conducted according to the principles expressed in the Declaration of Helsinki and approved by the Ethical Committee of Verona Hospital (Azienda Ospedaliera Universitaria Integrata, AOUI Verona, 2876CESC). Written informed consent was obtained from both parents before the start of the study.

### 2.6. Stool Sample Collection and Sequencing

Stool samples were collected (S1–S9, Figure 1) and stored at −20 °C until analysis.

DNA extraction and sequencing were performed at BMR Genomics S.r.l. (Padua, Italy). Briefly, the DNA was isolated with the Mobio Powerfecal kit (Mo Bio Laboratories, Inc., Carlsbad, CA, USA) adapted for the QIAcube HT extractor (Qiagen, Hilden, Germany). The V3–V4 regions of the 16S rRNA gene were amplified with previously described primers [11] and modified with the forward and reverse overhangs necessary for dual index library preparation (Illumina protocol) (https://web.uri.edu/gsc/files/16s-metagenomic-library-prep-guide-15044223-b.pdf, last accessed on 27 October 2020). The paired-end sequencing of the 16S rRNA gene amplicons was performed using the MiSeq Illumina platform (dual-indexing approach, 2 × 300 bp) (Illumina, San Diego, CA, USA). A mock community was included as a control.

Next to the metabarcoding analysis, a low coverage whole metagenome analysis (WMS) was carried out on five stool samples (S1, S2, S5, S6, S9): following the DNA isolation with the QIAamp cador Pathogen Mini Kit (Qiagen, Hilden, Germany) combined with the Mobio Powerfecal kit (Mo Bio Laboratories, Inc., Carlsbad, CA, USA), the shotgun libraries were prepared with Nexter Flex (Illumina, San Diego, CA, USA) then low coverage WMS was conducted with the MiSeq Illumina platform (dual-indexing approach, 2 × 300 bp) (Illumina, San Diego, CA, USA). The resulting output from both sequencing runs was a set of raw files in FASTQ format. All the reads have been submitted to the SRA archive and are available under the bioproject n. PRJNA746509.

### 2.7. Bioinformatic Analysis

Metabarcoding samples were analyzed using DADA2 (v1.18) [12], a tool that implements an error correction model and allows the identification of exact sample sequences that differ as little as a single nucleotide. The samples used in this study were sequenced at different times and came from different sequencing runs. As different runs can have different error profiles, the error rates for each run were determined individually. The final output of DADA2 was an amplicon sequence variant (ASV) table, which contains the count number of each ASV in each sample. DADA2 was run as described in https://benjjneb.github.io/dada2/bigdata.html (last accessed on 2 August 2021).

First, a preprocessing step was applied to the raw FASTQ sequences in order to remove reads containing ambiguous bases (Ns) and to trim the primer sequences using the Cutadapt program [13]. Next, the function filterAndTrim from the DADA2 package was used to remove low-quality bases at the end of reads setting the option truncLen to 290 and 250 for the forward and reverse read, respectively. The removeBimeraDenovo function was used to remove chimeras via the consensus method, and then the collapseNoMismatch function collapsed together all the reads that are identical up to shifts of length variation. Finally, taxonomic assignment was performed using the naïve Bayesian classifier method implemented in DADA2 (assignTaxonomy and addSpecies functions) using the Silva 16S database as reference (Version 138). A phylogenetic tree of the ASVs was obtained using the function AlignSeq implemented in DECIPHER (v2.16.1) [14], an R package to create multiple sequence alignments. The phylogenetic tree was obtained using the phangorn R package [15].

Whole metagenome reads were preprocessed to remove adapter sequences and trim low-quality bases using the Trimmomatic tool [16]. Taxonomy classification was performed using Kraken 2 software (The Center for Computational Biology, Johns Hopkins University, Baltimore, MD, USA), setting the option “confidence” to 0.5, against a database that includes all the prokaryotic, archaea, virus, and fungi downloaded from RefSeq. The taxa quantification produced by Kraken [17,18] was then further improved using Bracken [18], a highly accurate statistical method that computes the abundance of species in DNA sequences from a metagenomics sample.

The estimation of abundances of taxonomical groups obtained from both metabarcoding 16S and whole metagenome samples were visualized using the Krona tool [19].

Alpha diversity analysis was performed using the “estimate_richness” function implemented in the phyloseq package [20] and estimated using the Chao1 metric. Beta diversity analysis was conducted based on the Bray–Curtis distance metric using the “plot_ordination” function after performing rarefaction normalization on the count matrix.

## 3. Results

### 3.1. Clinical and Biochemical Outcomes

During the monitoring, the patient had no decompensation episodes, and clinical and neurological findings were remarkably well. Stool consistency and frequency were comparable to those expected at the same age. The mother reported colic episodes and more loose stools during metronidazole administration. Growth charts showed good weight increase but less pronounced linear growth. As for biochemical data (Table 1), ammonia levels and bicarbonate showed a steady course; in particular, metabolic acidosis crisis and high ammonia were never observed. MCA, a specific marker of disease [21], was high at diagnosis, but never showed spiking levels thereafter. Propionylcarnitine (C3) decreased from diagnosis levels and maintained a steady-state; conversely, glycine levels (another well-known disease marker) showed a slight increase.

### 3.2. Dietary and Nutritional Issues

At 4 days of life, free maternal milk was temporarily stopped, and a protein-free formula was administered for 24 h. Then, proteins were gradually reintroduced with controlled amounts of maternal milk and normal infant formula. During the first 4 months of age, since the maternal milk was not enough to cover the protein intake, a special formula was prepared to combine the maternal milk together with a normal infant formula (Nidina 1^®^ Nestlè, Vevey, Switzerland) and a protein-free formula (Basic P^®^ Milupa Nutricia, Danone, Paris, France). The infant formula Nidina 1^®^ contained *Limosilactobacillus* (former *Lactobacillus*) *reuteri* DSM 17,938 and 2′-O-fucosyllactose (2′-FL).

From 4 months of age, only normal infant formula and protein-free formula were used for the special diet. The mean dietary composition during the first 6 months of age was protein 4.7%, fat 53.3%, and carbohydrates 42% of total energy (%TE).

A nasogastric tube was been used in case the feed was not completed and for emergency regimen. Percutaneous gastrostomy was placed at 5 months of age (PEG in Figure 1). Fasting time was limited to 3–4 h during the day with continuous nocturnal feeding to avoid catabolism and lipid oxidation.

From birth to the first month of age, growth was lower than the normal z-score for the infant’s age but not indicative of malnutrition. During the following months, the weight increased, reaching the normal z-score, while the height growth slowed down to a −1.97 z-score at 8 months of age.

### 3.3. Urine Analysis

The first two urine samples were collected one week before (U1) and one day after (U2) the first metronidazole treatment, respectively, while U3 was collected during the second antibiotic treatment (Figure 1). As shown in the radar plot in Figure 2 (and in Appendix A), urine smell-prints in U1 and U2 were similar, except in sensors S7, S13, S18–20, and S30; U3 revealed marked distinctness when compared with samples U1 and U2; while in U4, collected at 5 months after six metronidazole treatments, the probiotic supplementation and the PEG placement, had a similar pattern of U1 and U2.

### 3.4. Microbiota Data

#### 3.4.1. Alpha- and Beta- Diversity Analysis Shows the Impact of Antibiotic Treatment

A general increase of the alpha-diversity was observed between S1 and S9 (Chao1 index), which suggested that the gut microbiota was evolving towards a higher complexity associated with the natural growth of the infant (from birth to 4 months of life) (Appendix A).

PCoA analysis was performed to calculate the beta-diversity using the Bray–Curtis distance. As reported in Figure 3, samples were distributed along Axis 1 according to the collection time (from S1 to S9): samples S1–S4 were related to the gut microbiota assessment at the first month of life and during the first metronidazole cycle, while samples S5–S9 were related to the analysis at 4 months of life, at the fifth therapy cycle, and with probiotic supplementation. The probiotic effect was not detected as the evolution of the gut microbiota is mainly due to its natural maturation from birth to 4 months, as also observed in the alpha-diversity analysis. The perturbation given by the antibiotic treatment was observed along Axis 2, mainly between S1 and S5 (collected before the therapy) compared to S2 and S6 (collected after the end of the therapy), respectively.

Alpha- and beta-diversity analysis calculated from metabarcoding sequencing data were also confirmed by the same analysis performed on the low coverage whole metagenome sequencing data obtained from samples S1, S2, S5, S6, and S9 (Appendix A).

#### 3.4.2. Microbiota Assessment at First Month of Life Unravels *Bac. fragilis* as Propionate Producer

The first stool sample (S1) was collected at two weeks of life (Figure 1), a week before the first treatment with metronidazole. As reported in Figure 4, the metabarcoding analysis showed that the gut microbiota of the patient was characterized by few bacterial populations, and it was dominated by the presence of *Bacteroides fragilis* (65% of relative abundance) (phylum *Bacteroidetes*)*,* followed by *Bifidobacterium breve* (25%) (phylum *Actinobacteria*). *Firmicutes* covered 7% of the total population, with *Veillonella* (family *Veillonellaceae*) and *Enterococcus* spp. (family *Enterococcaeae*) as the most represented (4 and 3% of the total, respectively), while *Proteobacteria* spp. (i.e., *Enterobacteriaceae* spp.) covered the last 3%.

The therapy with metronidazole was associated with a sharp removal of the *Bac. fragilis* population in the second stool sample (S2) collected one week after the end of antibiotic administration. The gut microbiota was still characterized by the presence of a higher abundance (32%) of *Enterobacteriaceae* spp. (where *Enterobacter* spp. represented 9% of all) and *Bif. breve* (32%), followed by *Enterococcus* spp. (which abundance increased until 30% compared to S1), and small fractions of *Streptococcaceae* and *Lactobacillaceae* spp. (phylum *Firmicutes*) (4%).

After 10 days of antibiotic treatment (S3), the gut microbiota was dominated by *Bif. breve* (46%) followed by *Proteobacteria* members, in particular *Enterobacteriaceae* spp. (28% of the total, with *Enterobacter* at 9%) and a smaller fraction of *Yersiniaceae* (*Serratia* spp., 5% of all). Interestingly, the population of *Bac*. *fragilis* increased again (20% of all). *Firmicutes* were mainly represented by *Lactobacillales* spp. (1%).

The *Bif*. *breve* fraction was the most abundant (56% relative abundance) in the stool sample collected 15 days after the end of the therapy (S4), while the abundance of *Bac*. *fragilis* slightly decreased to 9% of the total bacterial population. Generally, the gut microbiota was characterized by a higher complexity compared to S1, and it included members of family *Enterobacteriaceae* (26%), (mainly belonging to genera *Escherichia*/*Shigella* -7%-, *Klebsiella* -4%-, *Citrobacter* -3%), *Yersiniaceae* (*Serratia*, 2%) *Enterococcaceae* (*Enterococcus*, 8%), and *Streptococcaceae* (*Streptococcus*, 3%).

A low coverage whole metagenome analysis (WMS) was carried out on samples S1 and S2 in order to explore the gut microbiome structure down to the species level (Appendix A). Generally, data obtained via this sequencing strategy were consistent with the metabarcoding analysis, with only small changes in the relative abundances of the most represented populations.

#### 3.4.3. Gut Microbiota Was Characterized by *Bifidobacterium* spp. and *Ruminococcus gnavus* after Probiotic Supplementation

The gut microbiota composition was assessed again at 4 months of life, after four cycles of antibiotic therapy. Compared to the previous analysis, the microbiome was investigated when the patient received the fifth antibiotic treatment followed by supplementation of 300 mg/day of a probiotic formulation produced on purpose and composed of three bifidobacterial species (*Bifidobacterium longum* 04-DSM 23233, *Bifidobacterium bifidum* BB01-DSM 22892, and *Bifidobacterium breve* BR03-DSM 16604).

The sample S5 was collected right before the administration of the fifth antibiotic treatment (Figure 5). The microbiota was dominated by *Actinobacteria* with *Bif*. *breve* (64%), and *Eggerthella lenta* (1%) while *Bac*. *fragilis* was not detected, in contrast with what was observed in S1. *Enterobacteriaceae* species constituted 19% of the total bacterial population, mainly represented by *Escherichia* (7%) and *Klebsiella* (11%). Finally, members of *Firmicutes* covered the last 15%, including *Streptococcus* (6%), *Ruminococcus gnavus* (family *Lachnospiraceae*) (5%), and *Eubacterium* (family *Eubacteriaceae*) (2%).

Sample S6 corresponded to the end of the antibiotic therapy and to the first day of probiotic supplementation. The *Enterobacteriaceae* level was the same as S5 (19%); as for *Actinobacteria* relative abundance, *Bif*. *breve* was stable at 58% while *Actinomyces* spp. appeared reaching 1% of all. In the *Firmicutes* fraction (21% of the total), a higher abundance of *Enterococcus* spp. was detected (16%) followed by *Streptococcus* (3%), in agreement with the data obtained with sample S2, collected right after the end of the first antibiotic therapy (except for the presence of *Actinomyces*, not detected in S2).

A week after the end of antibiotic administration and after 6 days of probiotic supplementation (S7), *Bifidobacterium* spp. prevailed with a relative abundance of 74% of the total population. Among them, *Bif*. *breve* was still the dominant species (67%), followed by *Bif*. *bifidum* (4%) and *Bif*. *longum* (3%). *Enterobacteriaceae* species represented 11% of the total microorganisms, with *Escherichia*/*Shigella* (7%) and *Klebsiella* (2%) the most abundant. *Enterococcus* and *Streptococcus* spp. remained the most representative for phylum *Firmicutes* (11 and 2%, respectively).

The gut microbiota structure of sample S8, collected after two weeks of probiotic supplementation (S8), was comparable with S7. The main differences were related to *Bifidobacterium* spp., as *Bif*. *bifidum* and *Bif*. *longum* relative abundances decreased (0.7% and 0.6%, respectively), and to the *Firmicutes* members, which were now mostly represented by *Streptococcus* (6%) and *Ruminococcus gnavus* (3%). Sample S9 was collected after 3 weeks of probiotic administration, and it was characterized by a stable *Bifidobacterium* spp. population (*Bif. breve* at 52%, *B*. *longum* at 0.8%, and *Bif*. *bifidum* at 0.3%) and a higher abundance of *Ruminococcus gnavus*, which reached 22% of the total bacterial population. The abundances of other *Firmicutes,* as well as the *Enterobacteriaceae* fraction, were similar to those observed in the previous samples.

The low coverage WMS data obtained for samples S5, S6, and S9 were comparable with the outputs of the metabarcoding sequencing (Appendix A).

## 4. Discussion

Diet modification and medications reported in international guidelines are the main instruments to control and stabilize PA; however, few data are available related to organic acidurias (such as PA). Therefore, it is very difficult to analyze the role of the single components of the adopted treatment strategy given the extreme rarity of the disease and the different levels of its expression [4,22]. The growth rate is very important for children, in particular for those affected by inherited metabolic diseases under special diets. The patient in the present study showed good clinical conditions and outcomes during the observation period (from 0 to 8 months); as well as his neurological development and growth rate in the following months (he is 18 months old now). Despite his positive weight growth rate, his height growth was lower compared to standard rates, despite the intake of natural proteins and energy prescribed on a monthly basis was similar to those recommended by FAO/WHO/UNU 2007 for children of this age. Growth impairment in PA has been described as a complication of the disease [3]. A retrospective study on 55 PA patients from 16 metabolic centers [23] reported a smaller height rate from the age of 3 months to 10 years (height SDS −0.21 to −1.19 at different ages) compared to population standards, despite the fact that most of these patients also received an amino acid mixture, free of precursors of propionyl-CoA. In a European survey reporting the dietary management of 186 patients with PA from 47 European centers, different practices emerged on protein limitation and the supplementation of precursor-free amino acids, with a possible over restriction in natural protein [24]. Recently, an association between excessive natural protein intake above healthy recommended daily allowances and acute metabolic decompensation has been described in PA early-onset patients, confirming that targeting dietary protein amounts is extremely important for these patients [25].

Even though PA management protocols, if promptly adopted, are able to support normal growth and prevent episodes of metabolic decompensation, long term outcomes are still unsatisfactory [26]. In this perspective, the gut microbiota modulation, as part of a comprehensive dietary management approach, represents a promising target for the knowledge-driven development of new strategies in PA management. It is well known, in fact, that the microbial fermentation of non-digestible carbohydrates in the human large intestine provides short-chain fatty acids (SCFA) such as acetate, butyrate, and propionate [26]; more specifically, the relative contribution of the microbiota in the total propionate production in PA patients has been estimated to be 25% (with amino acids and lipids at 50% and 25%, respectively) [27]. Furthermore, the composition and metabolic activity of gut microbiota are often associated with constipation, which usually precedes decompensation and metabolic instability in PA [28]. Indeed, gut microbiota has already been addressed in PA management through the empirical use of oral antibiotics since they seemed to be effective in reducing the levels of propionate in the urine and plasma of PA patients. However, no data are available regarding their specific ability/mechanism in reducing the intestinal microbiota responsible for propionate production [26,29].

For this reason, in the present study, the gut microbiota composition of the infant diagnosed with PA was investigated through targeted 16S rRNA gene and low coverage whole metagenome sequencing (WMS). The two techniques provided consistent information related to the population structure as well as to the relative abundance of the microbial communities proving that they are interchangeable for the analysis of low complex microbiomes (as those of infants). Even though the WMS gave more detailed taxonomic resolution reaching the species level, 16S RNA gene sequencing was selected as the method of choice as it was less expensive, and the data analysis was faster [30].

In terms of complexity, the microbiota structure described during the first month of life (before starting the antibiotic treatment) was comparable with the normal neonatal gut, which hosts a relatively simple community of bacteria [31]. As for the phylogenetic composition, the gut microbiota of the infant diagnosed with PA was dominated by the *Bacteroides fragilis* population (phylum *Bacteroidetes*), which is usually present at a lower level in the microbiota of vaginally delivered and breastfed infants [32]; on the other side, bifidobacteria, mainly represented by *Bifidobacterium breve* (which is usually the most frequent species in newborns) [33], covered only 25% of the total bacterial population, while generally most shotgun and 16S rRNA V4 sequences (around 75%) in one-year-old babies map to members of the *Bifidobacterium* genus [34].

The Gram-negative *Bacteroides fragilis* is a common, anaerobic, non-spore-forming, bile-resistant commensal of the adult human gut; it is most frequently found on the mucosal surface, where it can contribute to the development and maturation of the host immune system [35]. This species adapts to environmental changes, and it is able to degrade a vast array of complex polysaccharides (including mucins and dietary fibers) to produce SCFA, mainly propionate, through the succinate pathway via methylmalonyl-CoA, the most abundant pathway in *Bacteroidetes* [36]. Propionic acid is usually absorbed and utilized by the host as an energy source and exerts a variety of distinct physiological health-promoting effects (i.e., anti-lipogenic, cholesterol-lowering, anti-inflammatory, and anti-carcinogenic activities) [37]; however, in PA cases, due to the disfunction of human PCC genes, propionic acid and its metabolites (i.e., propionyl-CoA) are built up to toxic levels with damages on the brain and nervous system. To date, this is the first report of a PA case in which the most likely propionate-producing population has been identified at the species level.

It has been suggested that the therapy with the antibiotic metronidazole is the most effective to reduce propionate-producing anaerobic bacteria in the gut microbiota [4,26,29]. According to the European Committee on Antimicrobial Susceptibility Testing (EUCAST, eucast.org), metronidazole (a member of the nitroimidazole class) is active against strictly anaerobic bacteria and protozoa, and its mechanism of action is DNA oxidation which leads to strand breaks and cell death. Data obtained in the present study confirmed the efficacy of metronidazole since the fraction of strict anaerobic bacteria, mainly composed by *Bac*. *fragilis*, was eradicated after a one-week antibiotic treatment, favoring other facultative anaerobic bacteria (i.e., *Enterobacteriaceae* spp.). *Bac*. *fragilis* emerged again at a lower level between 2–3 weeks after the end of the therapy, but then it was not detected at 4 months of life after four cycles of antibiotic treatment. Regarding other strict anaerobes, this antibiotic was not active towards *Bifidobacterium* spp. (mainly consisting of *Bif*. *breve*), which soon became the most abundant population after the therapy. This agrees with previous works reporting that metronidazole resistance is an intrinsic feature of bifidobacteria [38,39]. Besides metronidazole, other antibiotics have been proven useful (alone or in combination with metronidazole) against *Bac*. *fragilis*, such as beta-lactams (with or without beta-lactamase inhibitors), carbapenems, clindamycin, fluoroquinolones [37], but generally, their use is challenged by the emergence of resistance mechanisms (i.e., the presence of RND efflux pumps conferring multidrug resistance [37]). Even if the presence of *nim* genes coding for 5-nitroimidazole reductase and conferring resistance to metronidazole have been detected [38], a recent survey assessing trends and impact in antimicrobial resistance among *Bacteroides* and *Parabacteroides* spp. revealed that 226 *B. fragilis* strains (isolated from hospitalised patients between 2007 and 2017) were all susceptible to metronidazole with a MIC range between 0.016 and 0.47 mg/L, and no one evolved antibiotic resistance [40].

Although it is necessary for PA management, early-life antibiotic exposure (also combined with formula feeding instead of breastfeeding) is a disrupting factor for the gut microbiota homeostasis: compared to a healthy infant core microbiota (usually composed by species belonging to genera *Bifidobacterium*, *Clostridium*, *Enterococcus*, *Lactobacillus*, *Ruminococcus* and *Prevotella*), in this patient the treatment with metronidazole led to a higher abundance of *Enterobacteriaceae* spp. (phylum *Proteobacteria*), a population usually associated with metabolic disorders such as childhood overweight and obesity [41]. Furthermore, the presence of members of genera *Escherichia* and *Klebsiella* (usually associated to diarrhea; [42] could be related to the episodes of gut discomfort reported in the young patient following the antibiotic therapy. This hypothesis needs to be confirmed through the detection of endotoxins in the fecal water and/or via transcriptomics analysis.

The microbiota composition at 4 months of life was still dominated by *Bif*. *Breve,* whose abundance even increased following the supplementation of the probiotic formulation. The probiotic formulation was produced on purpose and included *Bif. longum* 04-DSM 23233, *Bif. bifidum* BB01-DSM 22892, and *Bif. breve* BR03-DSM 16604. Strains of these species were already widely used as probiotics in newborns and infants due to their high abundance in the GIT tract of these individuals, their capability of colonizing the gut and their beneficial effects in treating metabolic disorders, such as infantile colics and diarrhea [33].

The lower consumption of breast milk stopped at 4 months of age, and the use of an infant formula containing a source of *Limosilactobacillus reuteri* DSM 17,938 and 2′-FL, initially in combination with maternal milk and then as the main source of protein of the diet, surely affected the growth and numerousness of bifidobacteria in the gut of the young patient. In fact, human milk impacts gut microbiota mainly due to its human milk oligosaccharides and endogenous bifidobacteria [43]; furthermore, it has been observed that infant formulas enriched of 2′-FL showed initial evidence on promoting *Bifidobacterium* spp. in the gut microbiota of children [44].

Besides the dominance of the *Bif*. *breve* population (probably composed of both autochthonous as well as supplemented strains), the microbiota structure at 4 months after four cycles of antibiotics featured the presence of *Ruminococcus gnavus* (family *Lachnospiraceae*). This species had a low relative abundance before the therapy with metronidazole, and it was not detected right after it; it then increased again during the probiotic supplementation, becoming the second most abundant population after *Bif*. *breve*. As described by Moore et al. in 1976, *Ruminococcus gnavus* is an obligately anaerobic, non-spore-forming, Gram-positive commensal found in the intestinal tract of 90% of adults [32]. A previous study reported the susceptibility of this organism to metronidazole as a strict anaerobe, thus explaining the reason why it was not detected right after the antibiotic treatment [45].

Although *R. gnavus* was found to increase in pathological conditions, such as inflammatory bowel disease [46], it can be predominant in the infant’s gut as bifidobacteria, independently by the type of feeding or delivering. As a matter of fact, ruminococci belonging to *Lachnospiraceae* family have the same pathway of bifidobacteria for complex sugar and mucin degradation and they release metabolites that could be important for the evolution of a more mature microbiota [32,47]. At a functional level, it has been suggested that *R*. *gnavus* might promote protein synthesis and lean body mass formation, preventing amino acid oxidation; furthermore, data obtained from murine models indicated this organism could improve malnutrition and metabolic abnormalities [48]. Regarding the propionate production, this species was reported to release acetate and formate in PYG medium, but it was unable to produce propionic acid [49]; however, the genome sequence of *R. gnavus* ATCC 29149^T^ isolated from human feces harbors the gene coding for the propionaldehyde dehydrogenase (*pduP*) responsible for the conversion of propionaldehyde to propionyl-CoA, the precursor of propionate [36]. Nevertheless, it has been observed that propionate production via this pathway (known as propanediol pathway) was dependent on the carbohydrate available to growth, as *Blautia obeum* (another member of *Lachnospiraceae*) started to produce propionate from fucose and rhamnose fermentation, two carbohydrates known to be propiogenic [36,50,51]. Fucose is one of the five monosaccharides of HMOs naturally present in maternal milk, in variable quantities considering high or low excretion mothers. Some infant formulas are enriched in fucose because of their prebiotic role on infant microbiota. Rhamnose is part of heteropolisaccarides fibers such as pectin present in agrumes, apples and other fruits, carrageenan, and agar mostly present in seaweed. These types of fiber are sometimes present in special low protein foods that represent a big part of the PA diet after weaning. As for *R. gnavus*, previous works related to microbiota analysis in metabolic diseases reported that this microorganism as well as other members of *Lachnospiraceae* (*Ruminococcus inulinivorans*, *Ruminococcus torques*, *Eubacterium halii,* and *B. obeum*) were able to produce propionate via the propanediol pathway, which entered the mitochondrial tricarboxylic acid cycle with a potential toxic effect [52,53].

Besides the combination of standard clinical data and the investigation of the gut microbiota, the patterns of urine volatile organic compounds through an electronic nose were also investigated as a further clinical methodology to employ to improve the monitoring of patients and the disease progression [54,55]. VOCs are produced and emanated from the body as the result of metabolic processes; they both derive from the host and from microbial fermentation, but also as a host response to pathological processes. Usually, VOCs are investigated through the use of different analytical platforms such as high-end gas chromatography-mass spectrometry (GC-MS); in the present study, the electronic nose was employed, which technology was based on an array of chemical sensors that change their electrical resistance resulting in a different signal that describes a “smell-print”. Samples collected during metronidazole treatment, probiotic assumption, and PEG placement were characterized by specific patterns that could be related to the healthy or pathological status of PA progression. The present study is the first one to our knowledge reporting the use of the electronic nose in PA; although this technique needs to be further optimized and validated also through the identification of VOCs associated with each sample, it represents a further promising strategy in PA management.

## 5. Conclusions

The present study is the first, to our knowledge, reporting an analysis of the gut microbiota structure and its modulation following antibiotic treatment and probiotic supplementation in a patient with PA. Putative propionate-producing microorganisms have been identified, and probiotic intervention, combined with standard antibiotic therapy, has proven to be a suitable approach to restore and maintain the bifidobacteria population in the gut microbiome, and to keep the propionate producers at low levels.

Furthermore, a better understanding of the microbiota structure could provide novel indications to improve the dietary approach and clarify the possible impact of fucose and rhamnose in the microbiota of PA patients as propiogenic substrates.

However, a major limitation of this pilot study must be acknowledged. Data have been obtained from the analysis of a single case due to the extremely low frequency of PA in the general population; thus, a wider investigation on a higher cohort of patients is advocated to get a deeper insight into the evolution of the microbial ecology in the gut. Although it is mainly exploratory, this analysis can set the basis for future advancements, also at a functional level. In this perspective, whole metagenome data can be further investigated to (i) better understand which genes are responsible for propionate production in *Bac. fragilis* and (ii) to distinguish endogenous and supplemented *Bif*. *breve* in order to better quantify the effect of probiotic supplementation.

A further insight in PA management which is directly linked to the gut microbiota investigation, comes from the impact of the gut-brain axis (GBA) as well as the gut-liver axis (GLA) in disease progression. Regarding the GBA, the enteric nervous system could play a specific role in the pathogenesis of the extrapyramidal neurological damage in PA as it shares several pathways with the central nervous system [26]; while for GLA, metabolites produced by the imbalanced microbiota during the pathogenesis may promote an altered immunity inflammation, oxidative stress and insulin resistance [22].

The inclusion of other ‘omics’ technologies beyond those DNA-based, such as transcriptomics, proteomics, and metabolomics, represents the next step in the investigation and management of PA as they will allow a deeper insight into the microbial metabolic capabilities and interactions in the gut ecosystem [26]. Unravelling the optimal path in this metabolic labyrinth will be extremely important not only to set up more targeted dietary strategies but also to improve new non-invasive techniques (especially for children or vulnerable patients) such as the use of electronic nose to check the propionate in urine and in other biological samples (i.e., stools) to eventually monitor the disease progression and validate interventions/treatments (i.e., dietary approach, role of probiotic supplementation in reducing or stabilizing PA by the gut microbiota).

## Figures and Tables

**Figure 1 microorganisms-09-02599-f001:**
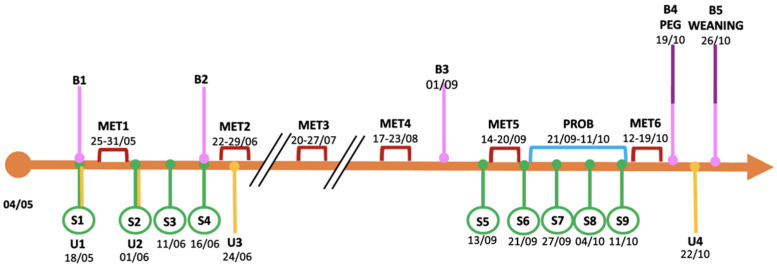
Experimental design of the study. B: blood sample; S: stool sample; U: urine sample; MET: metronidazole treatment; PROB: probiotic supplementation; PEG: percutaneous endoscopic gastrostomy. Months referred to year 2020.

**Figure 2 microorganisms-09-02599-f002:**
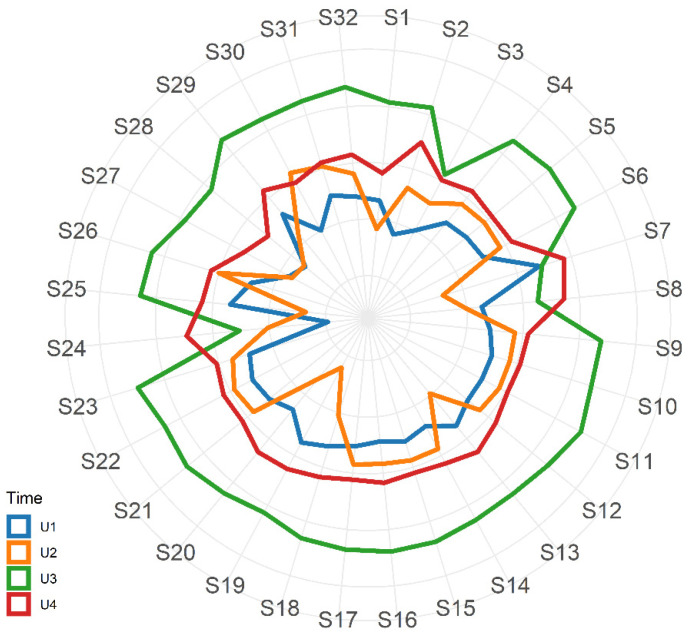
Radar plot with mean values of standardized responses from the 32 eNose nano sensors related to urine samples U1–U4.

**Figure 3 microorganisms-09-02599-f003:**
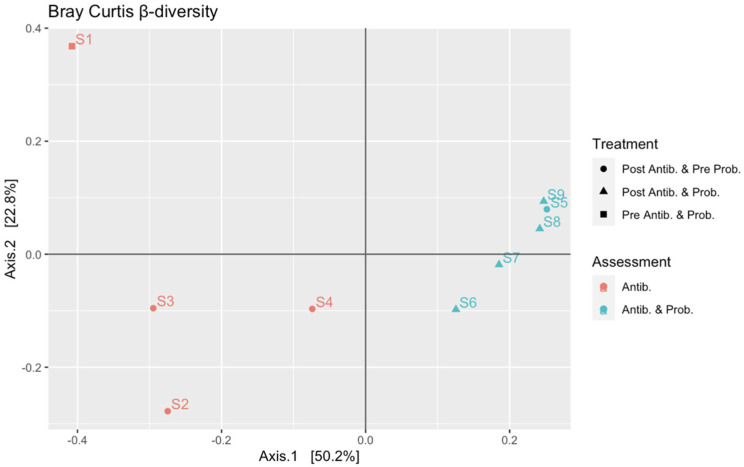
Beta diversity rarefaction based on Bray–Curtis distance metric using “plot_ordination”. Shapes indicate the type of treatment: circles are related to samples collected after antibiotic treatment and before the probiotic supplementation; triangles are related to samples collected after antibiotic treatment and probiotic supplementation; squares are related to samples collected before antibiotic treatment and probiotic supplementation. Colors indicate the two phases of the study: salmon orange is related to samples S1–S4 collected during the first month of life and associated with antibiotic treatment; turquoise is related to samples S5–S9 collected at 4 months of life and associated with antibiotic treatment and probiotic supplementation.

**Figure 4 microorganisms-09-02599-f004:**
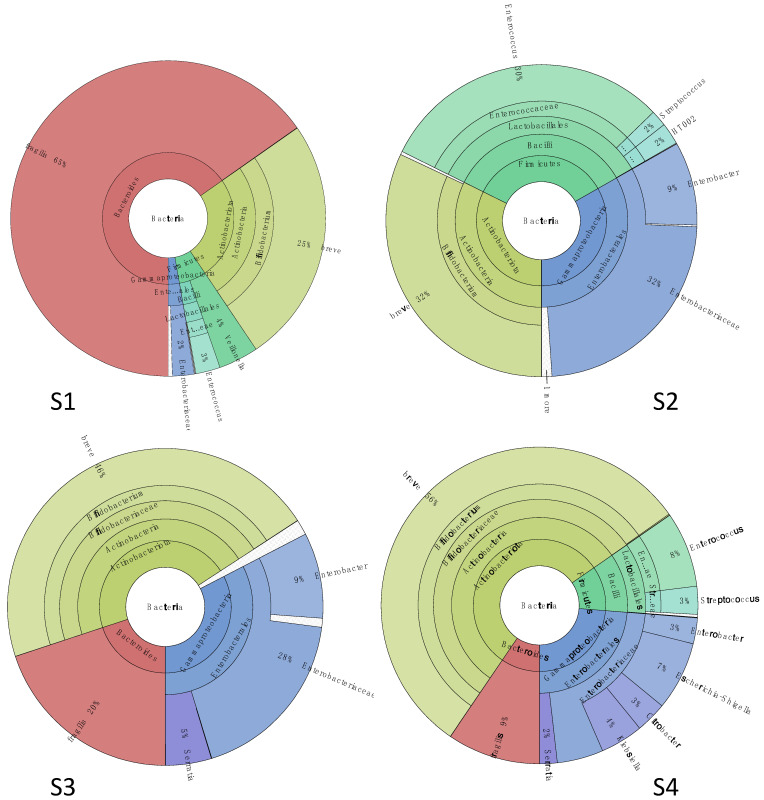
Microbiota structure assessed by metabarcoding analysis at first month of life. Sample (**S1**) was collected before the antibiotic therapy; samples (**S2**–**S4**) were collected at 1, 10, and 15 days after the end of the treatment.

**Figure 5 microorganisms-09-02599-f005:**
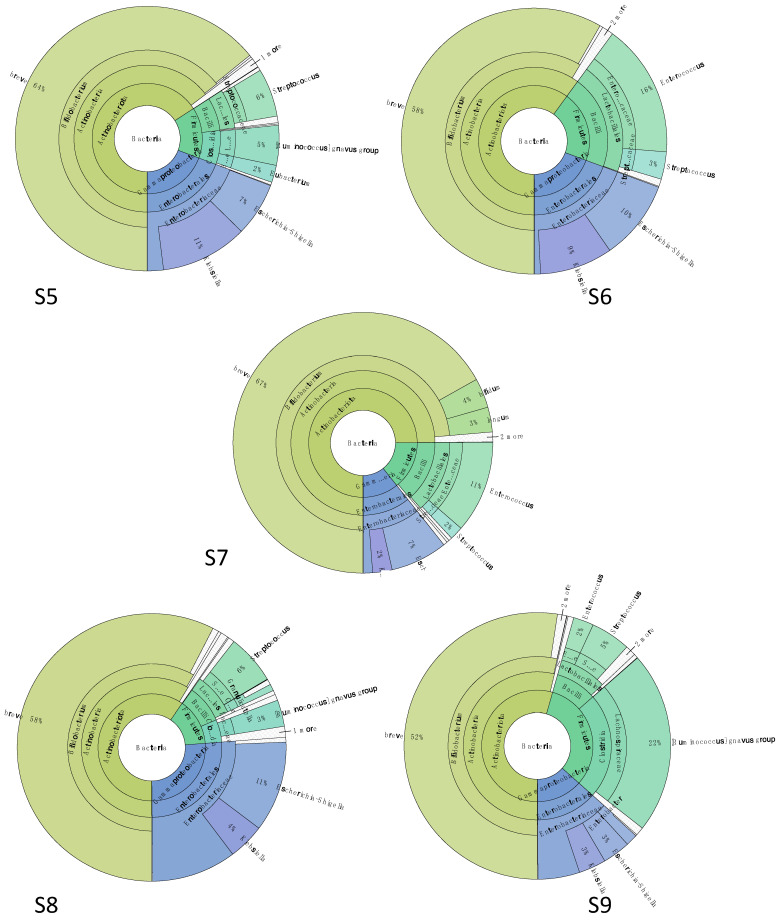
Microbiota structure assessed by metabarcoding data at 4 months of life. Samples (**S5**) was collected before the fifth antibiotic therapy, sample (**S6**) was collected 1 day after the end of fifth antibiotic therapy, and on the first day of probiotic supplementation, (**S7**) was collected 7 days after the end of 5th antibiotic therapy and on the sixth day of probiotic supplementation, (**S8**) was collected 14 days after the end of fifth antibiotic therapy and 13 days of probiotic supplementation, and (**S9**) was collected 21 days after the end of fifth antibiotic therapy and at the end of probiotic supplementation.

**Table 1 microorganisms-09-02599-t001:** Biochemical, growth, and nutritional data.

	B1	B2	B3	B4	B5	20 January 2021
Ammonia (µmol/L)	73	75	91	98	34	79
C0 plasma (µM)	57.58	59.52	30.55	35.59	49.92	48.61
C3 plasma (µM)	80.95	58.17	40.68	49.69	35.17	52.79
C3 DBS (µM)	59.38	31.99	20.63			
MCA DBS (µM)	37.5	12.9	16.8	17.4	7.6	11.3
Glycine plasma (µM)	939	429	768	1092	913	1030
protein (g/kg/die	1.52	1.16	1.19	1.36	1.18	1.14
Energy (kcal/kg)	131	128	88	90	82	83
Weight (kg)	3.016	3.950	6.450	7.5	7.6	8.8

DBS: dried blood spot; MCA: methylcitric acid. B1–B5 are reported in Figure 1; data in the last column is not reported in Figure 1.

## Data Availability

All the reads obtained by the sequencing platforms and related metadata were submitted to the Sequence Read Archive (SRA) of NCBI and are available under the bioproject n. PRJNA746509.

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
