# Peer review of "Assessing Gut Microbiota in an Infant with Congenital Propionic Acidemia before and after Probiotic Supplementation"

_microorganisms, 2021, doi:10.3390/microorganisms9122599_

Round 1

Reviewer 1 Report

The manuscript "Assessing gut microbiota in an infant with congenital propionic acidemia before and after probiotic supplementation" by Bordugo et al. describes a case report of a rare inherited metabolic disorder that leads to a toxic accumulation of propionic acid and its treatment through probiotics.

In the methods, the authors state "The taxa quantification produced by kraken was then further improved using [18], a highly accurate statistical method that computes the abundance of species in DNA sequences from a metagenomics sample."  Authors should state the name of the tool rather than just rely on the reference number.

Other than the above comment, the manuscript is in good condition.

Author Response

We thank the reviewer for the comment to the manuscript. We added the name of the tool (Bracken, line 231) as suggested.

Reviewer 2 Report

This manuscript corresponds to a case study, which has the interest of integrating the characterisation of the intestinal microbiota during the treatment of this infant. As indicated by the authors, it is hoped that this article will be a reference study, completed in time by the study of other cases. 
The study itself is very descriptive but well presented and discussed. Overall the study is well conducted.
One point of improvement would be the readability of the figures presenting the composition of the microbiota which is currently not very readable. 

Author Response

We thank the reviewer for the comment. 

We increased the font of the taxa names in Figure 4 and 5 to improve their readability as suggested.